# Validated Image Caption Rating Dataset

**Lothar D Narins**[1]    **Andrew Taylor Scott**[1]    **Aakash Gautam**[2]    **Anagha Kulkarni**[1]
**Mar Castanon**[1]    **Benjamin Kao**[1]    **Shasta Ihorn**[1]    **Yue-Ting Siu**[3]
**James Mason**[4]    **Alexander Mario Blum**[5]    **Ilmi Yoon**[1]
[1]San Francisco State University    [2]University of Pittsburgh
[3]Washington State School for the Blind    [4]UC Berkeley    [5]Stanford University
`{lnarins,bkao}@mail.sfsu.edu`
`{ats,ak,shasta,ilmi}@sfsu.edu`
`{marcastan4,alexander.m.blum}@gmail.com`
`aakash@pitt.edu, ting.siu@wssb.wa.gov, jmason888@berkeley.edu`

## Abstract

We present a new high-quality validated image caption rating (VICR) dataset. How well a caption fits an image can be difficult to assess due to the subjective nature of caption quality. How do we evaluate whether a caption is *good*? We generated a new dataset to help answer this question by using our new image caption rating system, which consists of a novel robust rating scale and gamified approach to gathering human ratings. We show that our approach is consistent and teachable. 113 participants were involved in generating the dataset, which is composed of 68,217 ratings among 15,646 image-caption pairs. Our new dataset has greater inter-rater agreement than the state of the art, and custom machine learning rating predictors that were trained on our dataset outperform previous metrics. We improve over Flickr8k-Expert in Kendall's $W$ by 12% and in Fleiss' $\kappa$ by 19%, and thus provide a new benchmark dataset for image caption rating.

## 1 Introduction

Image caption rating (ICR) is the task of estimating the quality of a caption for a given image. It is a growing area of research across various domains (e.g., [23, 31, 20, 32, 53]). ICR is becoming especially relevant since computer generated captions are used extensively for descriptive annotations. Identifying how good a caption is can be difficult because of the multiple aspects involved. At minimum, a good caption should correctly identify all relevant objects, where things are in space, the setting in which the objects are shown, and the interpretation of events depicted in the image [2]. Elements of subjectivity and value are inherent in it. Moreover, with the exponential growth in the use of images in the digital ecosystem, caption rating systems have to be scalable, requiring a mechanism that ensures consistency across a large scale.

However, there is a significant gap in the existing caption rating systems. First, while prior work has contributed multiple ICR datasets where human annotators were asked to assign quality ratings to image-caption pairs, they rely on ad hoc rating scales [33, 22, 50, 35]. While these datasets have been valuable in advancing the field and have been used extensively [55, 51, 4, 32, 44, 3], several of them suffer from high skew in the ratings with predominantly poor quality captions. Second, while some systems such as that by Cui et al. [9] and Levinboim et al. [33] provide a scale to rate generated captions, the granularity employed by these approaches are restricted to a simple binary scale (i.e., a caption is either good or bad). Indeed, the problem of image caption quality estimation has received substantial attention in recent years, underscoring the increasing need for reliable solutions [23, 31, 20, 32, 53].

37th Conference on Neural Information Processing Systems (NeurIPS 2023) Track on Datasets and Benchmarks.

A distinguishing characteristic of caption quality metrics is whether or not they depend on reference captions. Traditional NLP approaches such as BLEU, CIDEr, ROUGE, METEOR, and SPICE are monomodal, reference-based, and are unable to measure the nuance found in rich image captions and are constrained by the availability of references [41, 50, 11, 34, 3]. The success of reference-free approaches has opened new possibilities for ICR estimation in applications without access to reference captions [21, 47, 20, 7]. However, existing reference-free metrics lack rating granularity due to the use of oversimplified scales.

We sought to create a high-quality, validated dataset for image-caption rating. To this end, we developed a reliable and scalable data generation approach with a validated procedure. In this work, we contribute a new ICR dataset, generated through our approach. This approach employs a novel human-in-the-loop solution that has a rigorous human rater training procedure and a gamified data generation process with built-in quality control. The gamification aspect keeps the raters engaged and the built-in quality control steers raters toward higher quality answers. Our system is built on a 5-level image caption rating scale that captures subtle aspects of caption quality, as defined by accuracy and completeness. We demonstrate that the scale is teachable and enables consistency across raters. Our new dataset has greater inter-rater agreement than the state of the art, improving Kendall's $W$ by 12% and Fleiss' $\kappa$ by 19% over Flickr8k-Expert. Further, it improves Kendall's $\tau$ over Flickr8k-Expert in all reference-based and reference-free metrics we measured by 32% to 66%, and thus provides a new benchmark dataset for image caption rating.

## 2 Related Work: Image Caption Rating Scale and Datasets

Only a handful of ICR datasets exist [33, 10, 31, 50, 22]. Google Image Caption (GIC) [33] has 140k image-caption pairs and the Conceptual Caption Challenge dataset [10] has 5k image-caption pairs. Both of these datasets only used ratings on a binary scale ("good" or "bad") and score by the ratio of "good" to total ratings. This hinders their ability to capture incomplete or partially correct captions, or assess degrees of differences in caption quality (see Figure 1). In contrast, CapEval1k [31] uses a 5-level scale, but there is no definition for each level

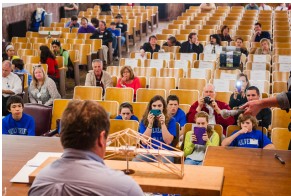 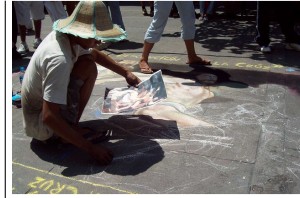

(a) Caption: "audience members at the conference hall"

(b) Caption: "street art on the sidewalk"

Figure 1: Two examples from GIC that received the highest possible rating (all 10 raters chose "good") even though salient aspects of the images are not captured, showing the limitations of a binary scale.

and the dataset only has 1,000 captions across 250 images. Another dataset, PASCAL50s [50], has 50k image-caption pairs across 1,000 images but the ratings are in a relative, pairwise and non-numeric scale, making it challenging to apply to our use case.

Flickr8k-Expert [22] is widely considered the state-of-the-art ICR dataset [55, 51, 3, 32, 20]. It has ratings on a graduated, numeric scale ranging from 1 to 4 with each rating associated with a specific meaning (see Table 1). Flickr8k-Expert has 5,822 captions across 1,000 images rated by 21 college students such that each caption has received 3 ratings. However, as seen in Table 1, the rating scales are underspecified and ambiguous

Table 1: Flickr8k-Expert rating scale.

| $r$ | Meaning |
| --- | --- |
| 4 | Describes the image without any errors. |
| 3 | Describes the image with minor errors. |
| 2 | Is somewhat related to the image. |
| 1 | Is unrelated to the image. |

leading to difficulty in consistent rating. In fact, from our analysis, we found that the Flickr8k-Expert dataset has fairly low inter-rater agreement (see Table 3). Moreover, the rating distribution in Flickr8k-Expert skews heavily toward ratings of 1 and 2, indicating lower caption quality (Figure 4a).

In summary, the major limitation across all the existing ICR datasets is that they have been generated using coarse or underspecified rating scales that are difficult for human raters to learn and apply consistently, leading to generally lower quality data. Our work fills this gap by providing a high-quality, validated image caption rating dataset that uses a scale that captures more meaning and nuance, enabling the raters to apply the scale more consistently.

# 3 Dataset Acquisition

## 3.1 Image Caption Rating Scale

Our 5-level scale embellishes previously deployed scales by addressing levels of context recognition (i.e. inferential information) in addition to accuracy and completeness of object recognition. Consideration for context has been a critical gap in evaluating and defining image caption quality. Our rating scale was developed in consultation with accessibility experts and informed by the Integrative Inferential Reasoning (IIR) framework, which describes a cognitive process for integrating local (explicit) knowledge with global (implicit, experiential, or world) knowledge [6]. IIR has been widely used in scholarship surrounding literacy [17, 29, 42], cognition [15, 48], and supporting neurodiverse populations [18, 40]. As applied to image captions, IIR offers a validated approach to measuring the quality of contextual information.

Table 2: Our robust image caption rating scale

| $r$ | Meaning |
| --- | --- |
| 5 | Objects, a general scene, and actions are correctly identified if present in the image. The caption describes what is seen and where things are in space. Interpretation of overall setting and/or event is included. |
| 4 | Objects and/or a general scene and/or an action are correctly identified but not every element is completely identified. The caption describes what is seen and where things are in space. There is no interpretation of an event. |
| 3 | Relevant objects are correctly identified. The caption describes what is seen but not where objects are in space. There is no description of the overall setting and no interpretation of an event. |
| 2 | Objects are partially correctly identified with some errors, but the caption is accurate enough to give an idea of what is happening in the image. The caption identifies most of the objects but might not identify everything. There is no interpretation of what anything means. |
| 1 | Objects are incorrectly identified. The caption gives the wrong idea about what is happening in the image. |

As shown in Table 2, our 5-level scale is designed to capture the extent of four essential aspects of image captions: (1) accuracy (e.g. 'objects are partially correctly identified'), (2) completeness (e.g. 'identifies most of the objects'), (3) local context (e.g. 'where things are in space'), and (4) global context and inferential information (e.g. 'interpretation of overall setting and/or event'). Global context also connects outside knowledge such as the audience for the captions (e.g. visually impaired and blind people). Our rating scale captures more nuance with less ambiguity than the state-of-the-art Flickr8k-Expert scale (contrast Table 1 and Table 2).

## 3.2 Gamified Data Generation

To promote human rater engagement, we frame the image caption rating task as a single-player, asynchronous, point-based game that is played in a web browser. We draw inspiration from the image labeling "ESP Game" [52]. In the ESP Game, two players try to guess the word the other is thinking for a given image. In our Rating Game, each individual player tries to predict the community consensus rating for given image-caption pairs in order to maximize their score.

We use a rigorous training procedure to align the raters' judgements to be consistent at following our scale. This, we believe, is one of the strengths of our dataset development process. This method is a form of Item Response Theory (IRT), which is considered the gold standard in the fields of education and psychology for scoring a given input [5]; for example, this is how the writing portions of the SAT and AP Exams are scored, ensuring similar scores from different reviewers. Our analysis in Section 4 demonstrates the validity of this approach.

**Human Rater Training**: Human raters must successfully complete a tutorial before contributing to our dataset. In the tutorial, the rater is guided through 10 image-caption pairs and asked to rate them. Their answer is compared with ground-truth ratings that were developed by multiple experts.

The human raters are given a score and feedback along with an explanation for why that rating was chosen by the experts. The score is given by $2 - |ground\_truth - rating|$. The point scores from the tutorial are not propagated into the game.

Raters who do not score at least 10 out of the maximum 20 points in the tutorial are put on "probation". They must rate 20 image-caption pairs from a curated set with a score of at least 25 before they can move on to the Rating Game. The image-caption pairs from the tutorial and from probation are not included in the final dataset.

**Rating Game**: Players in the game are presented with image-caption pairs and asked to select ratings from the 5-level scale, similar to the tutorial (Figure 2a). After a player submits their rating, they receive feedback and a score based on how their rating compares to the consensus rating of the other players so far (Figures 2b and 2c). For each image-caption pair, the player must wait at least 3 seconds before submitting their rating to discourage quick, random guessing. Players must score at least 25 points in the first 20 image-caption pairs or else their scores are reset and they are put on probation. The scoring algorithm, probation, and 3 second delay are intended to promote higher-quality ratings.

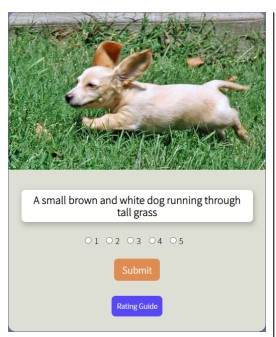

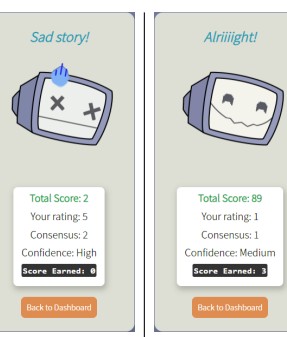

(a) Image, caption and 5-level scale. The meanings of the rating scale can be consulted anytime through the 'Rating Guide' button.

(b) Player feedback when the rating does not match the consensus.

(c) Player feedback when the rating matches the consensus.

Figure 2: Screenshots of the image caption rating game, showing the user interface and feedback.

**Scoring Algorithm**: The consensus rating, $r$, is computed by rounding the average of all the previous ratings for that image-caption pair. The score, $s$, assigned to the player models two intuitions: the first is that if the rating, $x$, is near the consensus the player should get a high score, and if the rating is far the player should get a low score. The second intuition is that if there is a high variance in the previous ratings the penalty for being far from the consensus should be lower, and if there is a low variance the penalty should be higher. These two intuitions are captured in Algorithm 1, which computes the player score, $s$, where $n$ is the total number of ratings available for the current image-caption pair (including the player's rating), $\sigma^2$ is the variance of the previous ratings, and $V_{max} = 4$ is the largest possible value of the variance. Since this scoring mechanism requires previous ratings, two initial ratings were primed with VSEPP [13] at first and from our own models later. These were replaced by human ratings as they were collected.

---

**Algorithm 1:** Player score, $s$.

$v \leftarrow 1 + \frac{1}{n}(1 + \frac{(n-1)\sigma^2}{V_{max}})$

$d \leftarrow \frac{|x-r|}{v}$

**if** $d \leq 0.25$ **then** $s \leftarrow 2$

**else if** $d \leq 0.5$ **then** $s \leftarrow 1$

**else if** $d \leq 1$ **then** $s \leftarrow 0$

**else if** $d \leq 1.75$ **then** $s \leftarrow -1$

**else** $s \leftarrow -2$

---

## 4 Validation

Before data collection, we conducted a user study to evaluate the efficacy of our rating scale and training procedure at generating high-quality ICR data. The user study, which was IRB approved, recruited 132 anonymized participants (college students at a 4-year public university). The user study simulated the Human Rater Training and the Rating Game that we used to generate our dataset (Section 3.2). For the simulated training procedure, participants needed to achieve at least 50% of the maximum possible training score in order to move on to the simulated Rating Game. Those participants who passed the training then rated 25 image-caption pairs in the simulated game. Five distinct images were used, and each image was paired five times with captions that represented each of the five levels of our rating scale. The image-caption pairs were presented in random order.

We analyzed rater competency by using the Partial Credit Model (PCM) [39], a technique designed to evaluate the quality of measurements obtained from ordinal scales. We applied the following rubric: When a user rated an item exactly as the ground-truth, we called this Exact Agreement (EA), and they received a score of 2. When they were off by one from the ground-truth, we called this Adjacent

Agreement (AA), and they received a score of 1. When a user rated an item that differed by more than one from the ground truth, we called this Lack of Agreement (LA), and they received a score of 0.

Following scholarship in Integrative Inferential Reasoning (IIR) [6], we analyzed the PCM data by using a Wright Map [54] and latent regression. The Wright Map allows us to visually compare the rater competence and the difficulty in rating image-caption pairs on the same scale. We use it to show that the rating scale can be consistently applied. The latent regression analysis allows us to see the strength of an explanatory relationship between our training procedure and rater competency. From the latent regression analysis, we see that the rating scale is teachable.

## 4.1 Wright Map

The Wright Map (Figure 3) shows the probability of raters to get EA or AA as a function of their competency after training. It also shows the difficulty of rating the captions in the simulated game. The vertical axis represents the rater competency in logits (also known as log-odds), seen in the third column. Logit scores of -1, 0, and 1 correspond to low, moderate, and high rater competency, respectively. The second column presents the histogram of rater competency which shows that the majority of the participants demonstrated moderate competency in the image-caption rating task. The first column overlays the results from the latent regression analysis that maps the rater's training score with their competency score. A training score of 50% (solid black line) was sufficient to perform moderately well during the Rating Game. The next section elaborates on this further.

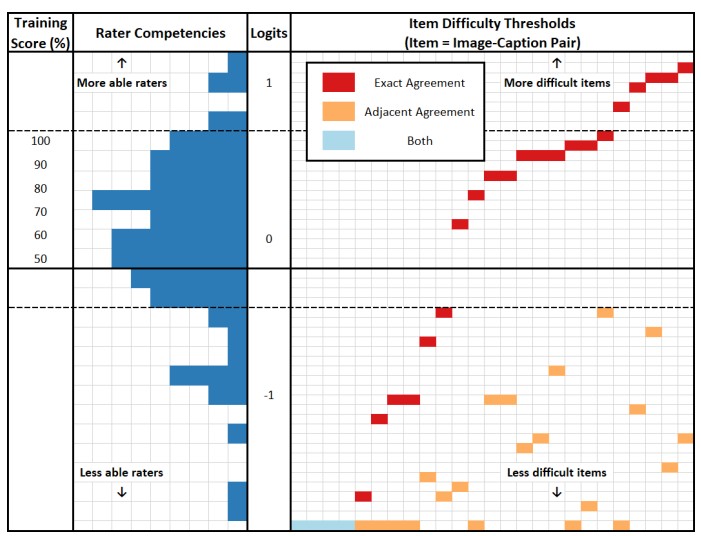

Figure 3: Wright Map augmented with predicted means from latent regression.

The right side of the Wright Map shows 25 image-caption pairs along the horizontal axis, with each column representing one pair. Each pair has two *difficulty thresholds*. The upper threshold, shown in red, marks the difficulty of achieving EA. The pairs have been sorted in ascending order of EA threshold. The EA threshold location on the vertical axis corresponds to the level of rater competence for which we expect a 50% chance of achieving EA. The second threshold, shown in orange, marks the difficulty of achieving at least AA with a 50% chance. For all 25 pairs, low rater competency is sufficient to have a 50% chance at AA. For the four pairs in the first four columns (denoted in blue bars), the two thresholds coincide suggesting that these four pairs are the easiest to rate in this collection.

As seen on the Wright Map, 103 out of 132 raters (i.e., 78%) are above all of the AA difficulty thresholds (lower dashed line); raters at that level can reliably achieve high agreement (EA or AA) on the simulated Rating Game image-caption pairs. The median rater competency was 0.01 logits. Raters that were more successfully trained (i.e. had higher competency scores) tend to achieve EA on more of the image-caption pairs; raters with a very high training score (100%, upper dashed line) tend to achieve EA on 80% of the pairs. Therefore, this leads us to conclude that people can apply our ICR scale consistently well.

## 4.2 Latent Regression

Latent regression was used to explain the relationship between training score and rater competency. It is presented in the first column of the Wright Map. The training score was a significant predictor of rater competency with a regression coefficient of 0.017, standard error of 0.003, and a $p$-value less

than 0.05. This means that each additional 10% training score is associated with a mean increase in rater competency of 0.17 logits.

The leftmost column of the Wright Map plots the training scores from 50% to 100%. Their placement along the vertical axis corresponds to the predicted mean rater competency score for each training score. At the training score of 50% (solid black line), the predicted mean rater competency is well above all the AA difficulty thresholds and above the EA difficulty threshold for 10 of the 25 image-caption pairs (lower dashed line). This suggests that, on average, even a rater with a training score of 50% is very likely to demonstrate at least AA on all pairs and has more than a 50% chance of EA on 10 of the pairs.

The significant finding from this analysis is that it provides external validity of rater competency in using our scale. This analysis also informs the choice of the minimum threshold for the training score of 50% which was used for rater selection during data generation (Section 3.2). The positive and significant regression coefficient of 0.017 indicates that raters who had higher training scores were better able to reach EA on more image-caption pairs; therefore, the scale is teachable.

## 5   Dataset Description

Our work contributes a new ICR dataset that we call VICR (Validated Image Caption Rating). The VICR dataset contains 9,990 images, 15,646 image-caption pairs, with a total of 68,217 ratings where each image-caption pair has between 3 and 7 ratings. Compared to the Flickr8k-Expert dataset, our VICR dataset has nearly 10 times the number of images, roughly 3 times the number of image-caption pairs, and approximately 4 times the number of ratings. Our ratings were generated through our Rating Game (see Section 3.2).

The images in the VICR dataset came from the MS-COCO 2014 validation set [35] and the Flickr8k-Expert dataset [22]. The captions were selected from five sources: (1) the original MS-COCO captions, (2) captions generated using the Pythia framework [24], (3) captions generated using the GLACNet model [27], (4) mismatched MS-COCO captions from other images, and (5) the original Flickr8k-Expert captions. The goal of sampling from these five sources was to create a balanced and wide variety of caption quality examples. The resulting captions have an average length of 10.9 words, where the shortest is 2 words, and the longest is 30 words.

113 participants generated the ratings for the dataset, earning about $24 per hour on average, depending on their score, and averaging 99 minutes of rating time. The participants took about 11 seconds on average to rate an image-caption pair. The data collection process was IRB approved, and participants had to sign a consent form before participating in the study.

Our dataset includes all of the image-caption pairs from the Flickr8k-Expert dataset but with ratings from our Rating Game. However, 2,350 image-caption pairs were held out from the game because they received all 1's in the Flickr8k-Expert ratings; these captions were effectively "unrelated to the image" (Table 1). We held these out to save time and money because they are known to be poor quality captions. We reincluded these held-out pairs in the final dataset. Furthermore, prior work (e.g., [50, 20]) excluded 158 pairs from the Flickr8k-Expert dataset because the captions appear in the references for those images; we also excluded those 158 pairs from our dataset.

## 6   Dataset Analysis

We present the analysis of our VICR dataset compared to the state-of-the-art Flickr8k-Expert dataset. A major advantage of our dataset is seen in the rating distributions (Figure 4), which show that VICR is much more balanced across the levels of caption quality. We apply common measures of inter-rater reliability, Kendall's $W$ and Fleiss' $\kappa$, as well as a novel approach using Kendall's $\tau$. We also demonstrate the usefulness of our dataset by training a reference-free machine-learning ICR predictor on it. This model outperforms all current reference-free and reference-based ICR metrics.

**Rating Distributions**: For comparative analysis, the rating distributions of the Flickr8k-Expert and VICR datasets are illustrated in Figure 4. For each image-caption pair, we use the rounded average of its ratings. Figure 4b shows that VICR has a more balanced rating distribution.

**Inter-Rater Agreement**: To determine inter-rater agreement we compute Kendall's $\tau$ [25], Kendall's $W$ [26], and Fleiss' $\kappa$ [14] on "virtual raters." We define a *virtual rater* to be a selection of one rating for each image-caption pair. Flickr8k-Expert only has 3 ratings per image-caption pair, which are sorted from lowest to highest. In order to compare our dataset with Flickr8k-Expert, we construct 3 virtual raters by randomly sampling 3 ratings for each image-caption pair and sorting those ratings such that virtual rater $X$ has the lowest rating, rater $Y$ has the middle rating, and $Z$ has the highest rating. We do this 100 times and take the average for each

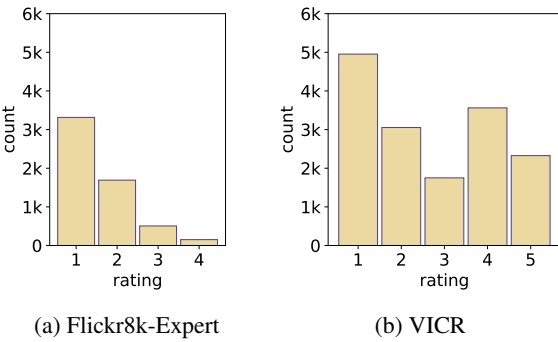

(a) Flickr8k-Expert  (b) VICR

Figure 4: Dataset rating distributions.

metric. In Table 3 we report these values scaled by 100 to be consistent with prior work [20]. For Kendall's $\tau$, we use "method $A$" for aggregation and variant $\tau_C$ to handle ties (see [20]). The standard deviations were between 0.08 and 0.12 for $\tau$, 0.05 for $W$, and 0.20 for $\kappa$.

The standard benchmark in previous work for evaluating ICR estimators is Kendall's $\tau$ coefficient [3, 32, 20]. The performance of state-of-the-art metrics in previous literature does not surpass a $\tau$ of 51.9 on Flickr8k-Expert [20], which we perceive to result from inherent limitations of the dataset itself. We test this by pitting the ratings of one rater against the rest as a proxy for human performance in the rating task. Therefore, we use Kendall's $\tau$ in a new way to measure inter-rater agreement. The first row of Table 3, $\tau_{X-YZ}$, shows the Kendall's $\tau$ correlation of virtual rater $X$'s ratings with the ratings of $Y$ and $Z$ as references. The other

Table 3: Inter-rater agreement.

|  | Flickr8k | VICR |
|---|---|---|
| $\tau_{X-YZ}$ | 47.7 | 75.6 |
| $\tau_{Y-XZ}$ | 54.8 | 77.8 |
| $\tau_{Z-XY}$ | 54.0 | 78.1 |
| $W$ | 84.0 | 93.8 |
| $\kappa$ | 48.8 | 58.1 |

rows are defined similarly. The resulting values from this analysis shows higher correlations from the VICR dataset, demonstrating greater inter-rater agreement.

The fourth row of Table 3, Kendall's $W$, shows how well the raters $X$, $Y$, and $Z$ all agree with one another. VICR has a Kendall's $W$ score that is 9.8 percentage points greater than Flickr8k-Expert, showing higher amount of inter-rater reliability.

Fleiss' $\kappa$ measures how well the virtual raters categorically agree with one another above random chance. We collapse categories for ratings 4 and 5 to all be categorical rating 4 in the last row of Table 3 because Fleiss' $\kappa$ is sensitive to the number of categories, and there are fewer categories in Flickr8k-Expert than there are in the VICR dataset. VICR has a Fleiss' $\kappa$ score that is 9.3 percentage points greater than Flickr8k-Expert, showing a higher amount of agreement within the virtual raters.

Another observation from Table 3 is that $\tau_{X-YZ}$ shows a lower correlation with the other virtual raters compared to $\tau_{Y-XZ}$ and $\tau_{Z-XY}$. This is a consequence of $X$ being the rater with the lowest rating per pair and the skew towards lower ratings. The effect appears much greater in the Flickr8k-Expert dataset because it has a much more severe skew.

**Evaluation With Baseline Rating Predictor**: To evaluate and study our datasets, we developed a baseline reference-free image-caption rating predictor (Figure 5). For the representation layer, the predictor takes a novel approach by employing ViL-

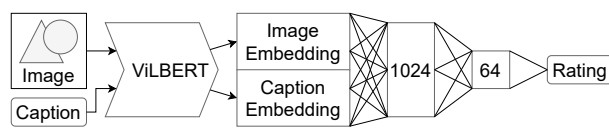

Figure 5: Architecture of our baseline rating predictor.

BERT [37, 38] co-attention embeddings, which were trained on the Conceptual Captions dataset [44]. Specifically, the input to the model (an image-caption pair) is represented by a 2048-dimensional vector created by concatenating the image and text embeddings from the final hidden layer of a pretrained ViLBERT model.

For the rating prediction, our model uses a dense neural network with two hidden layers. The first hidden layer has 1,024 neurons with a ReLU activation, and the second hidden layer has 64 neurons with a ReLU activation. The output layer consists of a single neuron, representing the predicted

rating. We use 80% dropout on both hidden layers. We use mean squared error for the loss. We trained for 4,000 epochs, with a batch size of 256, and with a learning rate of $10^{-5}$, decayed by 1% every 15 epochs. $Model_{Flickr8k}$ and $Model_{VICR}$ were trained on the Flickr8k-Expert dataset and the VICR dataset, respectively.

## 6.1 Experimental Results

We summarize the results of our experiments in Table 4 where we compare reference-based and reference-free metrics against our trained models and compare our VICR dataset with Flickr8k-Expert. Each dataset was split into 64% training, 16% validation, and 20% test. All measurements are of Kendall's $\tau$ scaled by 100 and were computed on the corresponding test sets. We used "method $A$" in aggregation and $\tau_C$ to be consistent with prior work [20]. Our models were each trained 5 times with different random seeds and the average Kendall's $\tau$ value is reported. The standard deviation of $Model_{Flickr8k}$ was 0.13 on Flickr8k-Expert and 0.20 on VICR, and the standard deviation of $Model_{VICR}$ was 0.17 on Flickr8k-Expert and 0.03 on VICR.

All of the metrics in Table 4 are significantly higher on our VICR dataset than on Flickr8k-Expert. This indicates that VICR is a more consistent and balanced dataset that can be modeled more easily than Flickr8k-Expert. Indeed, even $Model_{Flickr8k}$, which was trained on the Flickr8k-Expert dataset, gains a 34% improvement when predicting VICR ratings. $Model_{VICR}$ improves by 43% when tested on the VICR dataset as compared to Flickr8k-Expert.

Table 4: Kendall's $\tau$ correlation with ground-truth ratings on a test subset of each dataset for various metrics and predictors.

| Reference-based | Flickr8k | VICR |
|---|---|---|
| BLEU-1 | 33.7 | 55.8 |
| BLEU-4 | 31.6 | 51.8 |
| METEOR | 40.4 | 60.2 |
| ROUGE | 33.4 | 53.2 |
| CIDEr | 44.1 | 66.6 |
| SPICE | 41.2 | 60.4 |
| RefCLIPScore | 51.9 | 71.7 |
| ViLBERTScore | 50.1 | 66.9 |
| **Reference-free** | | |
| CLIPScore | 50.7 | 67.3 |
| VSEPP | 48.6 | 65.1 |
| VBAlignment | 49.9 | 65.8 |
| $Model_{Flickr8k}$ | 53.7 | 71.8 |
| $Model_{VICR}$ | 53.1 | 75.8 |

Our models, $Model_{Flickr8k}$ and $Model_{VICR}$, show the best performance over all metrics by a decent margin. $Model_{Flickr8k}$ slightly outperforms $Model_{VICR}$ on the Flickr8k-Expert dataset, while $Model_{VICR}$ significantly outperforms $Model_{Flickr8k}$ on the VICR dataset. We believe that $Model_{VICR}$ generalizes better because it was trained on a dataset with higher fidelity, more samples, and a more balanced ratings distribution. However, $Model_{Flickr8k}$ did better on the Flickr8k-Expert dataset, which we suspect is because the Flickr8k-Expert dataset is based on a 4-level scale, and skews toward lower quality, making it more difficult for $Model_{VICR}$ to predict.

A significant observation from comparing the Flickr8k columns of Tables 3 and 4 is that all of the metrics, including our best model, could not achieve a Kendall's $\tau$ of more than the best human rater, $\tau_{Y-XZ}$, who achieved 54.8 on Flickr8k-Expert. Indeed, it seems unreasonable to expect anything much higher from any metric, given the difficulty humans have in correlating the ratings themselves. Similarly, the metric scores on the VICR dataset are also capped below the human raters' highest score of 78.1 from Table 3. This again highlights the importance of a high-quality dataset.

The metrics we measured in Table 4 fall into two categories: reference-based and reference-free. The reference-based metrics are common NLP metrics and include RefCLIPScore [20] as well as ViLBERTScore [32]. RefCLIPScore is the reference-based version of CLIPScore and ViLBERTScore extends BERTScore [56] to the visual-linguistic domain by using ViLBERT embeddings. For ViLBERTScore, we used the fine-tuned model and reported the F1 metric (ViLBERTScore*$_F$). The reference-based metrics were calculated using the original reference captions from MS-COCO and Flickr8k, depending on the source of the image in question.

Besides our models, the other reference-free metrics are CLIPScore [20], VSEPP [13] and "VBAlignment." *VBAlignment* refers to the visual-linguistic alignment prediction task that is used in ViLBERT pretraining [37]. The outputs of this task are two logits representing "alignment" and "non-alignment." We apply softmax to the logits and use the "alignment" probability as the VBAlignment value for the image-caption pair. We use the pretrained model before fine-tuning on the 12 tasks from [38].

We used publicly available code for the reference-based metrics and CLIPScore from [19]; VSEPP from [12]; ViLBERTScore from [30]; and pretrained models and VBAlignment code from ViL-BERT [36]. Our model code is written in Python [49] using Keras [8] and Tensorflow [1]. The dataset and code for our model are available through our project website[1].

All experiments were run in-house on an NVIDIA RTX™ A6000 GPU server with AMD EPYC™ 7302 CPU and 256 Gigabytes of system memory. Our models took approximately 20 to 45 minutes to train. Extracting ViLBERT embeddings was the slowest part (approximately 930 milliseconds per image-caption embedding) mainly due to the image feature extraction.

## 7    Conclusion

Our work contributes to the growing argument on the importance of creating and maintaining high-quality datasets. Well-curated datasets have been shown to be strong competitors to large-scale alternatives [16, 43]; such datasets are smaller, more efficient, and less computationally demanding, requiring fewer resources to generate and manage. To this end, we present a high-quality and well-curated image caption rating dataset, called VICR, that was generated using a human-in-the-loop game and a validated five-level scale to facilitate image caption rating.

Our VICR dataset outperforms the state-of-the-art Flickr8k-Expert dataset in several important ways. VICR exhibits greater inter-rater agreement than Flickr8k-Expert, as well as higher correlation with reference-based and reference-free metrics. Additionally, machine learning models trained on VICR surpass state-of-the-art metrics when predicting ratings on VICR and Flickr8k-Expert. All of this suggests that our dataset is more consistent and balanced. Our approach, using a human-in-the-loop game and custom rating scale, can be extended to support other research groups in developing high-quality image caption rating datasets.

There are a few limitations of our work. The images in our dataset are limited in scope with regard to objects, actions, and locations, which constrains the applicability of the dataset. Flickr8k images have 6 categories [22] and MS-COCO images have 91 [35]. The captions are all fairly simple, without great variety in length and expression, and it may be difficult to generalize a model trained on them to evaluate more expressive captions. Another limitation of our work is that the images and captions are presented in our Rating Game without context; there is no surrounding information that gives the rater clues to the intent of the image and whether the caption accurately expresses that intent. The lack of such context hinders adaptation in cases such as improving accessibility where context has been shown to aid in image descriptions [28, 45, 46]. Our well-curated dataset and data collection methodology serve as stepping stones toward future research that incorporates contextual information. For example, we envision a version of our Rating Game that presents image-caption pairs in various contexts. A third limitation of our work is that paying the raters for data collection may have had an undue influence on the ratings by incentivizing them to prioritize monetary rewards over quality ratings. We would like to study this in future work by comparing paid raters with volunteers.

Our work holds the potential for significant social impact, both positive and negative. We anticipate that this work will lead to improved image and video caption quality, promoting accessibility and inclusivity in technology. However, we acknowledge that better image and video captions can empower bad actors in generating disinformation and manipulating public sentiment, potentially creating social unrest and decreased trust in visual media.

As part of future work, we would like to broaden the scope of images and create a wider variety of captions. Additionally, following the IIR framework, we would like to conduct a user study to further validate our scale against a simpler baseline. This will help to further quantify the current scale and will help with the development of ICR scales for other targeted applications.

We believe that our carefully curated high-quality VICR dataset enables opportunities to create consistent and reliable models that could automate the caption quality rating process. This provides new possibilities, not only for more accurate, rich, and descriptive image and video annotations, but also for searching through visual content such as for image and video retrieval. VICR gives the community a new benchmark dataset for image-caption rating.

---

[1] https://ai.youdescribe.org

## Qualitative Examples

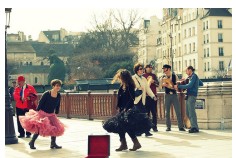

(a) A person descends a rope from a cliff into the ocean .
Ratings: 1 1 1 1 1

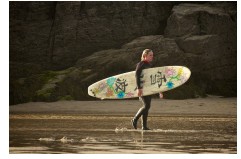

(b) a person is standing in the snow with a rope .
Ratings: 1 2 1 1 1

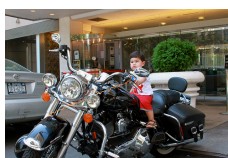

(c) A bridge filled with blue and white buses.
Ratings: 1 2 1 1

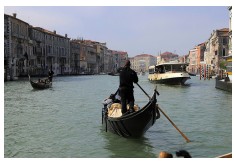

(d) A man and woman look back at the camera while standing in front of a red art structure .
Ratings: 1 1 1 1 2 2

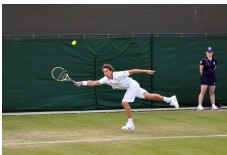

(e) Some kids are playing in a baseball game
Ratings: 1 1 2 3

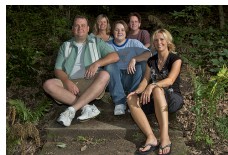

(f) girl posing on tree
Ratings: 2 3 2 1 2

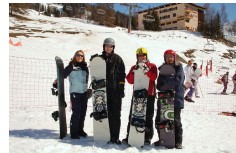

(g) a group of people are walking down a road .
Ratings: 2 2 3 1 3

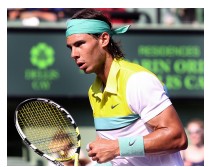

(h) A man without a shirt playing tennis .
Ratings: 1 4 2 2 3

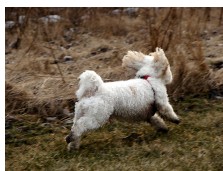

(i) Two white dogs running
Ratings: 3 3 3 3 2

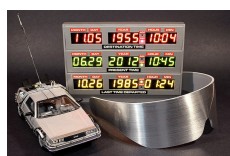

(j) a car is parked next to a clock
Ratings: 3 3 4 2

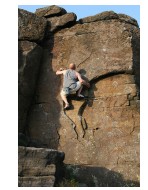

(k) A young man climbs a mountain , another follows below .
Ratings: 3 3 3 3 3

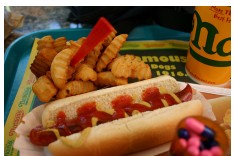

(l) a hot dog with ketchup and mustard on a table
Ratings: 3 4 4 4

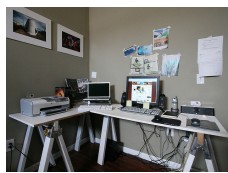

(m) a desk with two computers and a keyboard
Ratings: 4 4 4 4 3

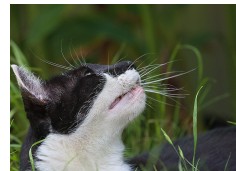

(n) The cat's gaze is focused on something above his head.
Ratings: 4 4 4 5 4

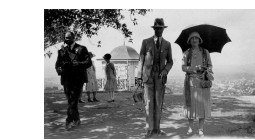

(o) A man in a suit is walking with a woman carrying an umbrella.
Ratings: 4 4 5 4

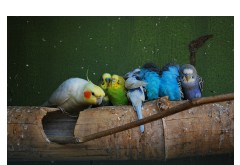

(p) a group of birds sitting on top of a wooden log
Ratings: 4 4 5 5

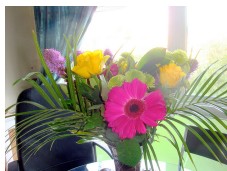

(q) A vase of fresh, colorful flowers on a table
Ratings: 4 5 5 5

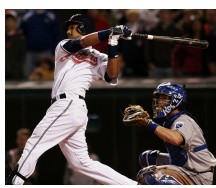

(r) A batter is swinging his bat while a catcher squats behind him.
Ratings: 5 5 4 5

(s) a bunch of vegetables on a counter top
Ratings: 5 5 5 5 5

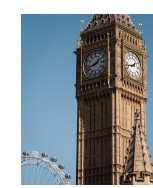

(t) the big ben clock tower towering over the city of london
Ratings: 5 5 5 5 5

Figure 6: A sampling of image-caption pairs from the VICR dataset with ratings.

## Acknowledgments and Disclosure of Funding

This work is part of the YouDescribe Project[2] in collaboration with Smith-Kettlewell Eye Research Institute[3], and made possible by funding from Ability Central[4] and a grant from the National Institute on Disability, Independent Living, and Rehabilitation Research (NIDILRR grant number 90 REGE0018).

Thanks to Charity Pitcher-Cooper, Brenna Gibson Tirumalashetty, Raya Farshad, Aditya Bodi, Abhishek Das, and Yash Kant for their invaluable contributions to this project.

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
