# OpenReview forum: "Validated Image Caption Rating Dataset"
_NeurIPS.cc/2023/Track/Datasets_and_Benchmarks — NeurIPS 2023 Datasets and Benchmarks Spotlight_

### Official Review · Reviewer_Qb6S · 2023-07-17
**Good contribution to context-free rating datasets (but I am hoping the field considers context-sensitive corpora going forward.)**

**Rating:** 7
**Confidence:** 4
**Clarity:** Yes.

**Strengths:**

The dataset represents a meaningful improvement over Flickr8K in terms
of context-free rating judgments: this corpus could support a number
of downstream tasks more effectively, e.g., evaluating evaluation
metrics, training evaluation metrics (that could act as, e.g.,
reinforcement learning reward). The experiments cover the corpus and
thoroughly make the case that indeed, this new corpus has higher
rating agreement than Flickr8K.

**Additional Feedback:**

Going forward, I'm hopeful that image caption dataset creation can
start incorporating contextual factors, because the quality of a
caption, e.g., for accessibility (as discussed in L103), depends on the
context and communicative intent of an image. Nonetheless, for
context-free ratings of this sort, this work represents a meaningful
advancement in the availability of a high IAA image caption rating
corpus.

**Correctness:**

Looks great, the authors methods are thorough. Caveat: I don't know too much about the methods described in S4, but it was interesting to learn about

**Documentation:**

Yes, the authors provide sufficient detail of the collection/curation process.

**Limitations:**

One limitation not discussed: the context-free nature of the setting. Considering alt-text generation as context-free is, in my view, an obstacle for alt-text generation.

**Opportunities For Improvement:**

I did have some concerns:

- The authors dataset is context-free, i.e., it simply pairs captions
  with images without a particular communicative intent. While other
  datasets also suffer from this shortcoming, I was a bit disappointed
  to see a new image caption rating dataset being collected void of
  context, which can often determine how good a caption is,
  particularly when the authors mention potential accessibility
  applications in L103.

- The authors mention Pascal50S's rating scheme as being inconsistent
  due to a "free-form, non-numeric scale" (L68). But, to my knowledge,
  evaluation on this corpus is conducted in a pairwise manner, i.e.,
  an eval metric is said to be better if it can reconstruct pairwise
  human judgments. What about the pairwise setup for that dataset?

- While I appreciate the qualification rounds the authors conducted,
  putting raters on "Probation" (L116) seems a bit harsh for simply
  disagreeing with the authors' ratings (unless most of the cases were
  not well-intentioned). Nonetheless, the authors do some additional
  analysis in sec 4 that suggests their rating scale is teachable
  through this process, and also got IRB approval.

**Relation To Prior Work:**

Yes, I don't see anything big missing that I'm aware of.

**Summary And Contributions:**

The authors collect a corpus of context-free (image, caption) ratings
spanning 10K images, 16K image/caption pairs, and 68K likert-style
ratings. Their data collection process involves a gamification that
rewards annotators for reproducing consensus ratings from prior
judgments; they show that their process results in higher
inter-annotator agreement compared to prior image/caption rating
corpora. They run experiments with existing metrics as well as trained
ViLBERT metrics, showing that all metrics achieve higher kendall tau
with VICR vs. Flickr8k.

---

> ### Author Response · Authors · 2023-08-23
>
> Thank you for taking the time to review our work and for your insightful comments and suggestions. We will respond to each point by point.
>
> **Opportunities For Improvement:**
>
> > - The authors dataset is context-free, i.e., it simply pairs captions with images without a particular communicative intent. While other datasets also suffer from this shortcoming, I was a bit disappointed to see a new image caption rating dataset being collected void of context, which can often determine how good a caption is, particularly when the authors mention potential accessibility applications in L103.
>
> Indeed, the captions were generated or selected without a particular context, but the scale was designed with a focus on accessibility for blind/visually impaired people, and hence the ratings are informed by that context.
>
> > - The authors mention Pascal50S's rating scheme as being inconsistent due to a "free-form, non-numeric scale" (L68). But, to my knowledge, evaluation on this corpus is conducted in a pairwise manner, i.e., an eval metric is said to be better if it can reconstruct pairwise human judgments. What about the pairwise setup for that dataset?
>
> Thank you for pointing out this error. We have corrected our writing in the manuscript.
>
> *Updated in manuscript:* Section 2 (67-68): “Another dataset, PASCAL50s [46], has 50k image-caption pairs across 1,000 images but the ratings are in a relative, pairwise and non-numeric scale, making it challenging to apply to our use case.”
>
> > - While I appreciate the qualification rounds the authors conducted, putting raters on "Probation" (L116) seems a bit harsh for simply disagreeing with the authors' ratings (unless most of the cases were not well-intentioned). Nonetheless, the authors do some additional analysis in sec 4 that suggests their rating scale is teachable through this process, and also got IRB approval.
>
> Yes, you are correct, this was intended to filter out not well-intended responses, where someone picked randomly or selected “3” all of the time.
>
> **Limitations:**
>
> > One limitation not discussed: the context-free nature of the setting. Considering alt-text generation as context-free is, in my view, an obstacle for alt-text generation.
>
> As earlier mentioned, we are not context-free: we are focused on accessibility for visually impaired and blind users though our work is applicable in a broader setting.

---

> > ### Comment · Reviewer_Qb6S · 2023-08-24
> > **confusions**
> >
> > Thanks for the response! I was confused a bit about:
> >
> > > but the scale was designed with a focus on accessibility for blind/visually impaired people, and hence the ratings are informed by that context.
> >
> > but are they? many papers suggest that context-free image descriptions are specifically *not* the most useful for low vision/blind folks. While I am in agreement with you that context-free descriptions are an OK starting point, I'd encourage the authors to engage more fully with the HCI work specifically related to low vision+blind users if they are going to use this as a motivation, e.g.,
> >
> > Context Matters for Image Descriptions for Accessibility: Challenges for Referenceless Evaluation Metrics
> > https://arxiv.org/abs/2205.10646
> >
> > "Person, Shoes, Tree. Is the Person Naked?" What People with Vision Impairments Want in Image Descriptions
> > https://dl.acm.org/doi/abs/10.1145/3313831.3376404
> >
> > > we are not context-free
> >
> > What I meant by context-free is: caption generation void of both communicative intent and broader image usage, e.g., on a website.

---

> > > ### Author Response · Authors · 2023-08-29
> > > **Added to limitations**
> > >
> > > Yes, you are correct, in that sense our dataset is indeed context-free. Thank you for clarifying and for those references. This has spurred a lot of discussion between the authors about context-aware *ratings*, and we intend to incorporate context in future data collections. We have added to our limitations in Section 7 and cited the references you provided.
> > >
> > > *Added to manuscript:* Section 7 (403-410): "Another limitation of our work is that the images and captions are presented in our Rating Game without context; there is no surrounding information that gives the rater clues to the intent of the image and whether the caption accurately expresses that intent. The lack of such context hinders adaptation in cases such as improving accessibility where context has been shown to aid in image descriptions [28, 45, 46]. Our well-curated dataset and data collection methodology serve as stepping stones toward future research that incorporates contextual information. For example, we envision a version of our Rating Game that presents image-caption pairs in various contexts."

---

### Official Review · Reviewer_ktRk · 2023-07-22
**Thoughtfully crafted rating system for image captions, inspired data labeling scheme, would have loved some qualitative examples**

**Rating:** 7
**Confidence:** 4
**Correctness:** Claims made in the submission look co…
**Clarity:** Yes, very much so.

**Strengths:**

The paper has many strengths including:
- thoughtful commentary on current state of rating/evaluation of image captions
- extensive detail on the worker pool / ethically sound and paid students under institutional review board
- new criteria based on many different  considerations
- an interesting gamified way of collecting high quality data for this task
- extensive statistics provided on annotator quality
- releases a dataset relevant to the domain

**Additional Feedback:**

N/A

**Documentation:**

Sufficient supporting detail provided.

**Ethics:**

I do not suspect there are ethical concerns with the submission / they call out the user population used and that things were approved via the IRB

**Limitations:**

I believe the authors addressed limitations of their work

**Opportunities For Improvement:**

Some things i would have liked to have gotten more details on:
- some qualitative examples of image/caption per rating value
- some commentary on how length and linguistic diversity impacted ratings of captions
- commentary on why game was used as opposed to some more mechanical turk like labeling force / what cost or speed implications that might have (and quality of course)
- commentary on how students were recruited to project
- commentary on if the automated rater was used in conjunction with the game
- details on how much time folks spend on task / per task
- commentary on distribution of caption sources, and ratings per source  distribution
- nit: any notes of bias or fairness concerns with the existing image or captions / impact on rating

further more i was curious if as folks played the game did they get better at rating, and if so how did you incorporate this information / consider their older ratings while they were learning

**Relation To Prior Work:**

Yes.

**Summary And Contributions:**

The VICR paper is a very well written paper that provides thoughtful commentary/critique on existing image caption rating/evaluation schemes, provides a well reasoned about set of new criteria for image captions, and outlines a gamified powered labeling scheme that produces high quality ratings as measured by many standard statistics for labelling. Further a dataset is released that builds upon 5 sources, and an automated rating predictor is described/provided.

---

> ### Author Response · Authors · 2023-08-23
>
> Thank you for taking the time to review our work and for your insightful comments and suggestions. We will respond to each point by point.
>
> **Opportunities For Improvement:**
>
> - We plan to add some qualitative examples in an appendix to our manuscript.
>
> - Yes, this is a limitation of our approach because we were constrained by the choices in Flickr8k captions (6 Flickr groups) and COCO captions (91 categories). Analysis of how length and linguistic diversity impacted ratings is a great addition to future work.
>
> - The purpose of the gamification was to provide an engaging context for learning. We could have used the game with Mechanical Turks but we had access to students in similar numbers which gave us an assurance of baseline quality. Although not reported in this paper, working with the students allowed us to conduct some qualitative surveys of how well the gamification worked for engagement.
>
> - After IRB approval, we solicited students' participation via departmental mailing list. We will add this information to the supplemental material.
>
> - The automated rater was only used to seed the ratings for the second data collection run, but not used anywhere after that.
>
> - Lines 252-255 in manuscript: “113 participants generated the ratings for the dataset, earning about $24 per hour on average, depending on their score, and averaging 99 minutes of rating time. The participants took about 11 seconds on average to rate an image-caption pair.”
>
> - We discuss that we tried to create a balance of quality in the captions from the distribution of caption sources, but we do not analyze how this affected ratings per distribution. This would be interesting to know and could be part of future work or added in appendix or supplemental material.
>
> - No fairness concerns, but as already noted, there is a bias in the categories of images depicted in Flickr8k and COCO. We are uncertain if this had an effect on the ratings and would like to study this further in future work.
>
> > further more i was curious if as folks played the game did they get better at rating, and if so how did you incorporate this information / consider their older ratings while they were learning
>
> Ratings from training and probation were not included in the dataset, but all ratings collected from the game were treated equally.

---

> > ### Comment · Reviewer_ktRk · 2023-08-29
> >
> > thanks for the response / details. i hope your submission gets accepted as this looks like a good paper. cheers.

---

### Official Review · Reviewer_Rnsb · 2023-07-24
**Solid dataset methodology and high quality dataset, but would be nice to see some more benchmarks and usability**

**Rating:** 6
**Confidence:** 4
**Correctness:** Yes, the claims and methods seem corr…
**Clarity:** The paper is very well written and ea…

**Strengths:**

- The paper is very clearly written.
- The rating scale is carefully and thoughtfully designed and seems reasonable. It takes into account multiple factors such as object completeness, local positions and global context of events.
- The entire human rating process is well designed and analysed, right from annotator training, gamification, validation (wright map)  and analysis. The gamification is a nice touch.
- A thorough analysis was performed to demonstrate that the rating scale is teachable. Annotators seemed to be treated well and fairly according to dataset creation standards.
- Given the proliferation of models trained on noisy image caption web data, the contribution of quality rating is timely and broadly relevant. This work seems to address a large gap in the research for this problem.


**Additional Feedback:**

Please see above for detailed feedback. I just had one clarification question:
 for the last 2 rows in Table 4, how is the model applied to the other dataset (not the one it is trained on)? Are all the trainable layers finetuned?


**Documentation:**

Yes.

**Ethics:**

Ethical concerns are discussed appropriately.

**Limitations:**

- The authors do a decent job of listing out limitations of the work. They say the images are limited in scope with regard to objects, actions and locations - it would be nice to discuss this more and maybe provide some details - what is the distribution of objects/categories/actions? This could be obtained automatically using some statistical analysis of the captions. It would also be nice to have statistics on the length/expressiveness of the captions. Apologies if these details are already provided somewhere in the supplementary.
- Can you expand on how paying the raters for data collection may have an influence on their ratings?
- One limitation that could also be mentioned is the time required to teach annotators the scale, however this also means that the scale is detailed and well-reasoned.
- The captions generated for the dataset are created using older models ([23], [26]). It would be better to augment the dataset with newer, more improved models (eg. GIT (https://arxiv.org/abs/2205.14100) , BLIPv2 (https://arxiv.org/abs/2301.12597) )


**Opportunities For Improvement:**

- What is missing here is some analysis of the usability of this dataset and model. For example, could the automatic captioning rating prediction model now be used to filter out poor quality captions from existing datasets, and then would a model retrained only on high quality captions do better than a model trained on all captions? Could the authors perform some preliminary experiments showing this with an off the shelf image captioning model? Or at the very least apply the trained model to another image captioning dataset in the wild, and qualitatively show that it can be used to filter out low quality captions?

- The dataset/model is only really compared to one existing ICR dataset, Flickr8k-Expert. It would be nice to show that models trained on VICR also perform favorably on other datasets with a domain gap, for example maybe on the Caption-Ext dataset (https://arxiv.org/pdf/1909.03396.pdf) , which has a slightly different scale? Or a more in depth analysis on why that dataset is not suitable. Is the usefulness axis also something worth looking into?

- While I agree having a fine-grained scale is more useful, the size of annotated images is smaller compared to larger datasets like [31] with a binary scale, which makes sense due to the large annotator time required. It would be interesting to show whether these datasets/annotations can be combined in a way to take advantage of the size/granularity trade off. Maybe with pretraining on the coarse scale and then finetuning on the 5-point scale? Or is there some way to show experimentally that this smaller, higher quality dataset is more useful than the large, coarse scale dataset? Maybe comparing with pretraining on both and then finetuning on the validation set of VICR, and then testing on the VICR test set.

- Only a single model (VilBERT) is trained on the dataset. Is it possible to ablate the model architecture and finetune another model, for example CLIP or similar?


**Relation To Prior Work:**

The paper compares to previous work well. it would be nice to have a few more experimental comparisons to other caption quality datasets, see above.

**Summary And Contributions:**

This paper produces a new image caption rating (VICR) dataset. It does this by first devising a 5-point well reasoned caption rating scale, which takes into account accuracy, completeness and context (both local and global). It then uses human annotators to annotate images from existing captioning dataset using a gamified interface. The entire human rating process is described thoroughly, right from annotator training, gamification, validation and analysis. The authors find that they newly designed rating scale is teachable. Finally the VICR dataset collected is compared to the FLICKR8K dataset using a number of reference-based and reference-free scores, including a trained model based on ViLBERT, and is shown to compare favourably.

---

> ### Author Response · Authors · 2023-08-23
>
> Thank you for taking the time to review our work and for your insightful comments and suggestions. We will respond to each section by section.
>
> **Opportunities For Improvement:**
>
> - The scope was to provide a dataset and validation study which opens up opportunities to build models like these. This suggestion is very much aligned with our current work and your suggestion gives us confidence to pursue this further in the future. The model presented in the paper was to establish a baseline for reference in future work.
>
> - Our focus is on a gradient scale, while the Caption-Ext dataset is based on binary ratings. Our stance has been that a binary scale cannot capture the nuance in image caption rating. Flickr8k-Expert is on a gradient scale that allows for a more direct comparison. As noted in lines 69-70 (section 2) of our manuscript, scholars and practitioners seem to consider Flickr-8k to be the current state of the art. Our focus in this paper is to present the VICR dataset and establish it as the new state-of-the-art for image caption rating.
>
> - Thank you. This is a great suggestion. We believe that a pre-training on a coarse scale before the 5-point scale could be helpful. We are going to incorporate this into our future work. However, as for this particular paper, we want to focus on presenting a validated dataset that can be used for a wide range of new opportunities. To this end, the models that we trained have a very specific function, which was to show how well our dataset performs in comparison to the state-of-the-art gradient-scale dataset (Flickr-8k).
>
> - Yes, we plan to try different embeddings in future work (in fact, we did try MS-GIT and CLIP embeddings but they did not show as much promise as ViLBERT embeddings), including training ViLBERT on more data to see if those embeddings offer better performance for VICR. BLIP-2 embeddings may also be useful but for this submission, we were mainly focused on the dataset generation, and the custom ML model we provide is just for baseline use.
>
> **Limitations:**
>
> - The objects, categories, and actions in our dataset are constrained by the limitations of the Flickr8k and MS-COCO datasets which are discussed in those papers. In brief, Flickr8k has 6 primary categories and COCO has 91 categories. The average length of a sentence in Flickr8k is 11.8 words. The average length in COCO is not reported. This resulted in some changes to our manuscript.
>
>   - *Added to manuscript:* Section 5 (250-251): “The resulting captions have an average length of 10.9 words, where the shortest is 2 words, and the longest is 30 words.”
>
>   - *Added to manuscript:* Section 7 (400-401): “Flickr8k images have 6 categories [22] and MS-COCO images have 91 [34].”
>
>   - We also added some of these statistics to the README in the supplemental materials.
>
> - Paying the raters may have incentivised participants to be more focused on their monetary reward than on the quality of the ratings. We would like to do a study in the future comparing paid raters with volunteer raters to see if this had an effect.
>
>   - *Added to manuscript:* Section 7 (404-405): “, by incentivizing them to prioritize monetary rewards over quality ratings. We would like to study this in future work by comparing paid raters with volunteers.”
>
> - That is correct: because the scale is detailed and well-reasoned, it takes time to train raters to use it. The Wright map analysis shows that the scale can be learned by going through training.
>
> - Yes, GIT and BLIPv2 were not available when we started this work. They would be great for generating captions for higher-valued ratings (ratings of 4 and 5), but for our purposes we used Pythia and GLACNet knowing that they generated slightly poorer quality ratings (ratings of 2 and 3), in order to obtain a more balanced distribution of caption quality in the resulting dataset.
>
> **Additional Feedback:**
>
> - Model-Flickr8k and Model-VICR are trained on training subsets of Flickr8k and VICR. They are then tested on each of the test subsets.
> ViLBERT is frozen but our model is a regression head on top. We take the embeddings from the frozen ViLBERT model and train a two-layer neural network to map embeddings to ratings.

---

### Official Review · Reviewer_CvUE · 2023-07-26
**Revision for Submission407**

**Rating:** 7
**Confidence:** 4
**Correctness:** No obvious correctness errors were fo…

**Strengths:**

+ The cardinality of the proposed dataset is an order of magnitude higher than the commonly used Flickr8k-Expert dataset for caption assessment.
+ The approach exploited to collect human judgments is novel.
+ The experimental results achieved considering both reference-based and reference-free methods demonstrate the consistency of the proposed VICR dataset.

**Additional Feedback:**

Please, see previous sections.

**Clarity:**

- I think I missed this detail. For reference-based methods what are the ground-truth captions for the VICR dataset?
- In Section 5, the authors state, "The captions were selected from five sources." However, no details about the selection criterion are provided. Furthermore, examining the dataset it is clear that some images have only one caption, while other images even have 8 captions. Authors should elaborate on this part and add a graph showing the number of captions per image.

**Documentation:**

It might be useful to report for each instance of the dataset the source from which the image and the caption were sampled.

**Ethics:**

I don't think there are any ethical issues with the presentation.

**Limitations:**

- Regarding the rating game, do not the authors think they are conditioning the users' judgments to be all the same or very similar? This misses the nuances of how each individual judges each image-caption pair based on his or her background, preference, etc. Are you sure that higher-quality ratings mean having equal ratings?
- There is not sufficient details pertaining to dataset documentation - as outlined in datasheets/artsheets paper. Please also refer to comments in the sections below. Please see detailed comments under ethics and documentation sections.

**Opportunities For Improvement:**

Considering the rounded average of each image-caption pair ratings as a ground-truth, the facets of the different judgments might be lost, wouldn't it make sense to model the distribution of judgments for each image-caption pair?

**Relation To Prior Work:**

The authors have effectively highlighted the limitations of the datasets currently used for caption ratings and their proposed dataset addresses those limitations.

**Summary And Contributions:**

In this work, the authors introduced a novel dataset called Validated Image Caption Rating (VICR). Evaluating the quality of image captions can be challenging due to its subjective nature. To address this, the dataset employs a robust rating scale and a gamified approach to collect human ratings. The dataset comprises 68,217 ratings given by 113 participants to 15,646 image-caption pairs. Compared to existing datasets, VICR demonstrates higher inter-rater agreement. Additionally, machine learning rating predictors trained on this dataset outperform previous metrics. As a result, the VICR dataset provides an improved benchmark for image caption rating.

---

> ### Author Response · Authors · 2023-08-23
>
> Thank you for taking the time to review our work, and for your insightful comments and suggestions. We will respond to each point by point.
>
> **Opportunities For Improvement:**
> > Considering the rounded average of each image-caption pair ratings as a ground-truth, the facets of the different judgments might be lost, wouldn't it make sense to model the distribution of judgments for each image-caption pair?
>
> In the final dataset, all of the individual ratings are provided. We use the rounded-average of ratings for each image-caption pair for the histograms in our Dataset Analysis (Section 6), and in the game, we use rounded-average ratings when calculating consensus. In the Dataset Analysis section, the histograms show the rounded-averages of ratings because we were focused on ensuring the quality of image-caption ratings were relatively evenly distributed. In the game, we used rounded-averages of ratings to provide a target to measure the performance of the rater. The count and variance of the previous ratings are also considered when calculating the score (see Algorithm 1). Also in the training and evaluation of the machine learning model, the individual ratings are used rather than the rounded average.
>
> **Limitations:**
> > - Regarding the rating game, do not the authors think they are conditioning the users' judgments to be all the same or very similar? This misses the nuances of how each individual judges each image-caption pair based on his or her background, preference, etc. Are you sure that higher-quality ratings mean having equal ratings?
>
> Yes, you are correct, we aligned users’ judgments to be similar. We believe this to be one of the strengths of our dataset development process. Before beginning data collection, we used the existing research on high-quality descriptions to develop a scoring rubric for assessing the quality of a given image-caption pair. We explicitly trained our raters to use this scoring rubric, assessed their capacity to do so, and only allowed individuals who demonstrated an ability to accurately apply this rubric to contribute ratings to our dataset. Thank you for drawing attention to this issue.
>
> *Added to manuscript:* Section 3.2 (110-115): “We use a rigorous training procedure to align the raters’ judgements to be consistent at following our scale. This, we believe, is one of the strengths of our dataset development process. This method is a form of Item Response Theory (IRT), which is considered the gold standard in the fields of education and psychology for scoring a given input [5]; for example, this is how the writing portions of the SAT and AP Exams are scored, ensuring similar scores from different reviewers. Our analysis in Section 4 demonstrates the validity of this approach.”
> > - There is not sufficient details pertaining to dataset documentation - as outlined in datasheets/artsheets paper. Please also refer to comments in the sections below. Please see detailed comments under ethics and documentation sections.
>
> We will add more information to the README in the supplementary materials concerning this. Could you please elaborate on specifically which material you are looking for? We are planning to answer the questions from *Datasheets for Datasets* (Gebru et al., 2021).
>
> **Clarity:**
> > - I think I missed this detail. For reference-based methods what are the ground-truth captions for the VICR dataset?
>
> The VICR dataset is composed of Flickr8k and MS-COCO images. For the reference-based methods, we used the reference captions from the respective datasets (not provided as part of our dataset). We have updated the manuscript to clarify this point.
>
> *Added to manuscript:* Section 6.1 (370-372):  “The reference-based metrics were calculated using the original reference captions from MS-COCO and Flickr8k, depending on the source of the image in question.”
> > - In Section 5, the authors state, "The captions were selected from five sources." However, no details about the selection criterion are provided. Furthermore, examining the dataset it is clear that some images have only one caption, while other images even have 8 captions. Authors should elaborate on this part and add a graph showing the number of captions per image.
>
> We prioritized creating a balanced distribution of ratings and so we gathered them from those 5 sources to create that balance. We randomly selected from the sources to fill 5 buckets of quality ratings to create that balance. These were then rated in the game by humans.
>
> The images coming from MS-COCO were given either 1 or 2 captions, and the images from Flickr8k have between 1 and 10 captions because that’s what was in that dataset. All in all, 80% of the images in the VICR dataset have 1 caption, 10% have 2 captions, and the remaining 10% have between 3 and 10 captions.
>
> Thank you for pointing this out, we will put this into the dataset documentation.

---

> > ### Comment · Reviewer_CvUE · 2023-08-30
> >
> > I would like to thank the authors for clear and timely responses to my comments. Regarding the details of the dataset I think it is more than comprehensive to answer the questions in Datasheets for Datasets (Gebru et al., 2021). I am happy to raise my score and wish you all well in the acceptance of the manuscript.

---

### Author Response · Authors · 2023-08-23

Thank you all for reviewing the paper and for the constructive suggestions. We are glad that all the reviewers saw value in our work. The feedback has helped us understand better how to clarify our presentation and in general how to establish a clearer line of argumentation.

We have considered the feedback carefully and have made corresponding changes to our manuscript. These changes include referring to prior work that we had missed, adding more documentation in our dataset description, and adding an appendix with qualitative results. We have also added individual comments where we share specific changes we have made and provide clarifications.

We are grateful for your kind and thoughtful feedback and for this opportunity to iterate on the draft.

---

> ### Author Response · Authors · 2023-08-26
> **Updates to supplementary materials: datasheet and qualitative examples**
>
> We have now added several items to our supplementary materials. We have included a new dataset documentation file called "Dataset Datasheet.pdf" following the format in Datasheets for Datasets (Gebru et al., 2021). We have added three qualitative visualization files in html format: VICR_visualize_25.html contains 25 image-caption pairs with ratings from our dataset, VICR_visualize_dataset_order.html contains all 15,646 pairs in the same order as in VICR.csv, and VICR_visualize_sorted.html contains all pairs sorted by average rating from lowest to highest. These html files are standalone and viewable through any web browser.
>
> Thank you, again, for the feedback.

---

### Decision · Program_Chairs · 2023-09-22

**Decision:**

Accept (Spotlight)

**Comment:**

The paper tries to address an interesting question and all the reviewers support the study. The ACs also share the same excitement with the reviewers about the paper.